# MADE: A Computational Tool for Predicting Vaccine Effectiveness for the Influenza A(H3N2) Virus Adapted to Embryonated Eggs

**DOI:** 10.3390/vaccines10060907

**Published:** 2022-06-06

**Authors:** Hui Chen, Junqiu Wang, Yunsong Liu, Ivy Quek Ee Ling, Chih Chuan Shih, Dafei Wu, Zhiyan Fu, Raphael Tze Chuen Lee, Miao Xu, Vincent T. Chow, Sebastian Maurer-Stroh, Da Zhou, Jianjun Liu, Weiwei Zhai

**Affiliations:** 1Human Genomics, Genome Institute of Singapore, Agency for Science, Technology and Research, Singapore 138672, Singapore; chenh1@gis.a-star.edu.sg; 2Key Laboratory of Zoological Systematics and Evolution, Institute of Zoology, Chinese Academy of Sciences, Beijing 100101, China; qiuqiu.wang@warwick.ac.uk (J.W.); liuyunsong@ioz.ac.cn (Y.L.); wudf@ioz.ac.cn (D.W.); 3School of Mathematical Science, Xiamen University, Xiamen 361005, China; 4University of the Chinese Academy of Sciences, Beijing 100049, China; 5Bioinformatics Core, Genome Institute of Singapore, Agency for Science, Technology and Research, Singapore 138672, Singapore; quekel@gis.a-star.edu.sg (I.Q.E.L.); shihcc@gis.a-star.edu.sg (C.C.S.); 6IHiS—Integrated Health Information Systems, Singapore 554910, Singapore; zhiyan.fu@ihis.com.sg; 7Bioinformatics Institute, Agency for Science, Technology and Research, Singapore 138671, Singapore; leetc@bii.a-star.edu.sg (R.T.C.L.); sebastianms@bii.a-star.edu.sg (S.M.-S.); 8State Key Laboratory of Oncology in South China, Collaborative Innovation Center for Cancer Medicine, Guangdong Key Laboratory of Nasopharyngeal Carcinoma Diagnosis and Therapy, Sun Yat-Sen University Cancer Center, Guangzhou 510060, China; xumiao@sysucc.org.cn; 9NUHS Infectious Diseases Translational Research Program, Department of Microbiology & Immunology, Yong Loo Lin School of Medicine, National University of Singapore, Singapore 117545, Singapore; micctk@nus.edu.sg; 10School of Biological Sciences (SBS), Nanyang Technological University (NTU), Singapore 637551, Singapore; 11National Public Health Laboratory (NPHL), Ministry of Health (MOH), Singapore 308442, Singapore; 12Department of Biological Sciences, National University of Singapore (NUS), Singapore 117543, Singapore; 13Department of Medicine, Yong Loo Lin School of Medicine, National University of Singapore, Singapore 117597, Singapore; 14Center for Excellence in Animal Evolution and Genetics, Chinese Academy of Sciences, Kunming 650223, China

**Keywords:** egg passage adaptation, vaccine effectiveness, influenza H3N2 virus, adaptive evolution, vaccine production

## Abstract

Seasonal Influenza H3N2 virus poses a great threat to public health, but its vaccine efficacy remains suboptimal. One critical step in influenza vaccine production is the viral passage in embryonated eggs. Recently, the strength of egg passage adaptation was found to be rapidly increasing with time driven by convergent evolution at a set of functionally important codons in the hemagglutinin (HA1). In this study, we aim to take advantage of the negative correlation between egg passage adaptation and vaccine effectiveness (VE) and develop a computational tool for selecting the best candidate vaccine virus (CVV) for vaccine production. Using a probabilistic approach known as mutational mapping, we characterized the pattern of sequence evolution driven by egg passage adaptation and developed a new metric known as the adaptive distance (AD) which measures the overall strength of egg passage adaptation. We found that AD is negatively correlated with the influenza H3N2 vaccine effectiveness (VE) and ~75% of the variability in VE can be explained by AD. Based on these findings, we developed a computational package that can Measure the Adaptive Distance and predict vaccine Effectiveness (MADE). MADE provides a powerful tool for the community to calibrate the effect of egg passage adaptation and select more reliable strains with minimum egg-passaged changes as the seasonal A/H3N2 influenza vaccine.

## 1. Introduction

As an RNA virus, influenza evolves rapidly and causes annual epidemics resulting in 3 to 5 million cases of severe illness, and 290,000 to 650,000 deaths [1]. Due to its rapid antigenic drift [2,3], the World Health Organization (WHO) organizes consultation meetings twice a year to recommend the best candidate vaccine viruses (CVVs) for the world. Despite many years of efforts, vaccine efficacy against influenza viruses, especially the H3N2 subtype, remains suboptimal [4,5,6].

There are many factors that can influence influenza vaccine effectiveness (VE) [7,8]. In addition to antigenic drift [2,3], several other factors ranging from glycosylation of the hemagglutinin [9], egg passage adaptation [10], repeated vaccination [11], imprinting and cohort effect [12] as well as the waning effect [13] can contribute to the variability of vaccine efficacies [4]. Among these factors, egg passage adaptation during vaccine production has been implicated in low vaccine efficacy across many years [14,15,16], but its link to vaccine effectiveness has been constrained to individual mutations occurred in different years, lacking a systematic metric integrating substitutions across many sites. Using a probabilistic inference procedure known as mutational mapping [17], our recent study found that egg passage adaptation was driven by substitutions in several key codons in HA1 across years [18] and the strength of egg passage adaptation has become progressively stronger in the recent past. When the H3N2 influenza had just crossed species boundary from birds to humans in 1968, passaging them in an avian environment led to weak positive selection across many codon positions. As influenza viruses became well-adapted to the human host, passaging them in embryonated eggs (an avian environment) led to much elevated passage adaptation at a set of functionally important codons (e.g., codon 186 and 194) [18]. The intensity of natural selection calculated as the rate of nonsynonymous to synonymous ratio (i.e., *d_N_*/*d_S_*) is often infinity due to the extremely high rate of nonsynonymous changes [18]. In order to measure the overall strength of egg passage adaptation, a new metric known as the adaptive distance (AD) was developed to quantify the intensity of egg passage adaptation in a given candidate vaccine virus (CVV) [10]. For the first time, the overall level of egg passage adaptation can be combined into a single metric (AD) integrating multiple substitutions distributed across many codons. Interestingly, a strong negative correlation was found between AD and VE for vaccine strains in recent years, and as much as 75% of the variation in VE can be explained by AD [10]. Thus, increasingly stronger egg passage adaptation has led to influenza vaccines with more substitutions and poor vaccine efficacy.

In this study, we extended the findings from our previous work [10,18] and have developed a new computational tool designated as MADE: Measuring Adaptive Distance and predicting vaccine Effectiveness using allelic barcodes (MADE). Based on allelic status (i.e., allelic barcodes) at a given set of positively selected codons in the HA1 gene for egg passage adaptation, MADE can: (a) calibrate the level of egg passage adaptation by calculating AD for any given candidate vaccine virus (CVV) and predict its potential vaccine effectiveness; (b) since there are a large number of sequences in the database with unknown passage history, and egg passage adaptation can confound many evolutionary analyses including “contaminating” the signal of adaptation in humans [18,19], MADE can infer whether a given isolate with unknown passage history has been grown in embryonated eggs using a machine learning approach known as XGBoost [20]. If the inferred passage history is not embryonated eggs, MADE can predict its passage history (e.g., MDCK or other growth medium) for the input sequence. In general, we aim to develop a computational tool that can select reliable strains with minimum egg-passage changes for the seasonal A/H3N2 influenza vaccine.

## 2. Materials and Methods

### 2.1. Data Curation and Computational Analysis

The steps retrieving the public data and performing the computational analysis are similar to our recently published work [10] and are explained in greater detail in the Appendix A. From the Global Initiative on Sharing All Influenza Data (GISAID) [21], we retrieved 76,489 influenza A/H3N2 HA1 sequences and their associated passage histories (Appendix A). After quality control and further annotating passage histories of all sequences (Appendix A), 69,362 sequences were retained for subsequent analysis and the numbering of different codon positions is based on the HA1 sequence with 329 amino acids. The subsequent analysis follows the below steps: (a) sequence alignment and phylogenetic reconstruction [22] (Appendix A); (b) using a probabilistic approach known as mutational mapping to sample possible evolutionary histories of the sequences [17]. The mutational mapping method is constructed based on theories from the continuous time Markov chain and has been used widely in inferring the history of mutations [17]. (c) Given the sampled evolutionary histories of the input sequences, two statistical tests (i.e., the enrichment and convergent test) were used to identify codons driving the egg passage adaptation [10] (Figure 1A). In the enrichment test, we asked the question whether substitutions happened in a given codon are more enriched in egg terminal branches, while in the convergent test, we asked the question whether substitutions happened in a given codon along the egg terminal branches are more likely to be convergent substitutions (Appendix A). Using theories from the continuous time Markov chain, we can explicitly test for the significance of these patterns across all codons and identify codons responsible for egg passage adaptation. (d) Calculating the enrichment score (ES) for all the alleles (amino acids) at the codons responsible for egg passage adaptation (Figure 1B). The ES is calculated as the ratio of frequencies for a given allele in egg-passaged strains versus all the strains (see Results (Section 3)). Alleles with high ES score are those alleles occur specifically in egg-passaged strains. For each sequence, we can extract a high-dimensional vector of ES scores at the codons responsible for egg passage adaptation. This high-dimensional vector serves as an allelic barcode for a given input sequence. (e) Given the high dimensional ES scores for all the sequences, we can perform principal component analysis across all strains. The adaptive distance (AD) is defined as the distance between the input strain (e.g., CVV) and the centroid of the major cluster for most of the non-egg sequences (Figure 1). Adaptive distance integrates patterns of adaptive evolution across multiple codons and captures the intensity of overall level of egg passage adaptation (see Results (Section 3)). (f) Performing the linear regression between AD and VE for the historical vaccine strains, and predicting the potential VE of the input CVV based on its AD value. Since the predicted VE is not a direct measurement of VE in humans, but rather is predicted based on the level of egg passage adaptation captured by AD, we denoted it as VE_ad_.

### 2.2. Classify Input Strains with Unknown Passage History

Since MADE can effectively measure the strength of egg passage adaptation in a given sequence, it also provides an additional feature distinguishing egg strains and non-egg-passaged strains. In order to achieve this goal, we first transformed the sequence dataset into the one-hot encoded dataset, based on which Random Forest and XGBoost methods were applied to predict the passage histories (see Appendix A for details). The performance of the algorithm was evaluated using the precision score (F1-score, defined as 2 × (Precision × Recall)/(Precision + Recall)), which measures the accuracy of a test based on precision and recall. Here, precision is the ratio of true positives to all predicted positives (i.e., TP/(TP + FP)), while recall is the ratio of true positives to all actual positives (i.e., TP/(TP + FN)).

## 3. Results

From the GISAID database, we extracted 69,362 HA1 sequences from the H3N2 influenza spanning 1968 to 2018 (Methods (Section 2), Appendix A). Among all the sequences, 898 sequences were passaged in embryonated eggs and majority of the sequences were grown in mammalian cells (e.g., MDCK cells, Appendix A). Using a maximum likelihood approach [22], we inferred the phylogenetic relationships and the mutational parameters (e.g., substitution matrix) of all the sequences (Appendix A). Conditioning on maximum likelihood parameters, we performed mutational mapping, a probabilistic approach to infer the history of mutations along each branch for all the codon positions [17,23]. In order to identify codons responsible for egg passage adaptation, we implemented two statistical tests based on patterns of sequence evolution [10]. In the enrichment test, a statistical procedure is implemented to examine whether changes along terminal branches were more enriched in egg-passaged sequences (i.e., egg terminal branches) than expected by chance. The second test (i.e., the convergent test) identifies codons with repeatedly the same change (i.e., convergent substitutions) along egg terminal branches (Methods (Section 2) and Appendix A). Applying these two statistical tests, we identified 17 positively selected codon positions in the HA1 gene at a false discovery rate of 5% (Figure 1A). These codons strongly enriched for antigenic epitopes B and D as well as the receptor-binding sites (RBS) (Figure 1A). For example, 8 out of 17 codons are from epitope B sites, which had 180 nonsynonymous substitutions and only 6 synonymous substitutions, indicating strong adaptive evolution driving the evolution of these codons in these functional domains (Appendix A). When we inferred substitutions along the egg terminal branches for these 17 codons, we observed an average of 289 nonsynonymous changes and 11 synonymous changes, which contributed to 63.5% of all the nonsynonymous changes, but only 5.2% of the synonymous substitutions for all codons along the terminal branches. This high level of nonsynonymous to synonymous ratio has hitherto not been observed in any naturally evolving system [24], indicating extremely strong adaptive evolution in embryonated eggs (Figure 1A).

When an allele (amino acid) is preferentially selected in egg-passaged sequences, it will be in relatively high frequency in the egg sequences relative to the background frequency. In order to systematically measure frequency differences in egg-passaged strains versus all the sequences, we previously developed an enrichment score (ES) as p_egg_/p_all_ for each allele observed over all 329 HA1 codons [10]. Here, p_egg_ denotes the proportion of strains carrying the specific allele in the egg-passaged strains, while p_all_ represents the proportion of strains carrying the specific allele among all sequences. Interestingly, a few alleles from several codons responsible for egg passage adaptation exhibit extremely high ESs (i.e., high frequency in egg strains, low frequency in the total set), such as 186V (ES = 35.13) and 194P (ES = 46.6) (Figure 1B). Alleles with high enrichment scores also tend to enrich in important antigenic epitopes such as epitope B as well as RBS (Appendix A). In summary, we defined an important statistic known as the ES score which could measure the preference of individual alleles at codons responsible for egg passage adaptation.

Given the extremely high level of nonsynonymous to synonymous change (*d_N_*/*d_S_* is often infinity), we previously developed an important metric known as the adaptive distance (AD) to measure the level of egg passage adaptation that occurred in the sequence [10]. For a given sequence, enrichment scores for alleles at the 17 HA1 positions provided a 17-dimensional metric defining the unique sequence feature of that input sequence (i.e., allelic barcode). Sequences bearing preferred alleles (i.e., amino acid) at selected codons will have very high ES scores across the 17 dimensions. Using principal component analysis (PCA), we projected the 17-dimensional space into the first two leading principal components (PCs). Interestingly, most of the sequences not passaged in embryonated eggs cluster as a major group (group 1, Figure 1C), whereas egg-passaged sequences reside within various clusters away from the major group. Inspecting allelic configurations in each dispersed cluster, specific egg-passage related alleles are enriched in the clusters similar to the antigenic map [3] (Figure 1C,E,F). We thus defined the adaptive distance (AD) as the distance from the input strain to the centroid of the major group representing viruses without egg passage adaptation (i.e., group 1, Figure 1C, Methods (Section 2)). When we performed a linear regression between AD and VE curated from a recent meta-analysis combining many studies [5], we found a strong negative relationship with a correlation coefficient of R^2^ = 0.741 (*p*-value = 0.039) (Figure 1D). This high linear correlation allowed us to predict VE of any input CVV with a regression line as VE_ad_ = −0.022 × AD + 0.78 (Figure 1D).

When we investigated the historical records for these vaccine strains, the egg passage adaptation correlated very well with the VE data. For example, A/Victoria/361/2011 was recommended as the vaccine strain for 2012–2013 by the WHO [14]. When we calculated AD for multiple A/Victoria/361/2011 strains grown in MDCK cells, the MDCK-passaged sequences yielded a mean AD of 0.384 (located within cluster 1, Figure 1C), whereas the egg-passaged sequences had a mean AD of 29.528 (located in cluster 3 in Figure 1C), resulting in a low predicted VE_ad_ (~12.1%). Looking at the substitutions in the egg-passaged vaccine strain, it carries three adaptive changes in the antigenic epitope B (H156Q, ES = 2.139, G186V, ES = 35.130) as well as epitope D (S219Y, ES = 5.186), consistent with the low vaccine performance in the 2012–2013 season [14]. These observations suggest that egg passage adaptation can often affect the antigenicity of the virus and subsequently lead to poor vaccine efficacy (see Discussions (Section 4)).

To make these methods available to the research community, we developed MADE (Figure 2), a software that can perform the above-mentioned functions for any given candidate CVV including (1) calculating ESs for alleles at the 17 positively selected codons in HA1 gene, (2) performing principal component analysis and calculating the AD for the CVV, and (3) predicting VE_ad_ of the input CVV based on the signal of adaptive evolution measured in AD. As MADE is assuming appreciable levels of egg passage adaptation, the prediction will not be performed for strains with very little signal of egg passage adaptation (Figure 2). Since many egg-passaged sequences bear specific alleles, we developed machine learning methods to predict whether the input sequence has truly been passaged in embryonated eggs. Using one-hot encoding along with machine learning methods such as Random Forest [25] and XGboost [20], we can predict the passage history of the input sequence very accurately (F1-score of 0.8624 under Random Forest and 0.864 under XGboost) based on the allelic configuration at these 17 codons.

In addition to predict potential vaccine efficacy of an egg-passaged CVV, as 1/3 of the sequences in the GISAID database have unknown passage history and egg passage adaptation can often confound many sequence studies including evolutionary analysis of influenza evolution in humans [18,19], the machine learning model in the pre-screening step can also be further extended to distinguish egg-passaged strains from non-egg strains. We thus constructed a multi-class classification method which can further classify the unknown passage history into four passage types including “Cell”, “MDCK”, “SIAT” and “Other”. These multi-class machine learning models can achieve good performances with F1-score of 0.9566 under the Random Forest and 0.9521 under the XGboost method (see Methods (Section 2) and Appendix A).

After all the analysis, MADE will output a complete report including: (a) the sequence information of the CVV; (b) allelic status and enrichment scores at the 17 codons responsible for the egg passage adaptation; (c) AD and predicted VE_ad_; (d) the likely passage history of the input sequence if the growth medium is not embryonated eggs. The backend engine for generating the report is “R Markdown”. MADE is freely available online in two different versions with an open-source form available at github (https://github.com/chenh1gis/MADE_docker_v1 (accessed on 1 June 2022)) and an interactive web interface available at http://39.105.1.41/made (accessed on 1 June 2022).

## 4. Discussion

Using a powerful probabilistic approach known as mutational mapping, we have developed an efficient tool that can measure the extent of egg passage adaptation in a given CVV and predict its potential vaccine effectiveness (VE_ad_). For the first time, the strength of egg passage adaptation can be integrated into a single metric (i.e., AD) alleviating the challenge studying different substitutions across years. The strong correlation between AD and VE provides an important connection linking egg passage adaptation to vaccine effectiveness. Following up statistical evidences provided by MADE, subsequent experimental methods (e.g., animal challenge experiments using ferrets) can be employed to test the immunogenicity and functional consequence of these passaged strains (e.g., analyzing elicited antibodies for their neutralizing abilities against the wild-type and passaged viruses) [26]. Moreover, the statistical approach developed here can potentially be further extended to other viral types including influenza H1N1. To facilitate its accessibility, we have integrated MADE with Flusurver and FluCluster-AI (https://flusurver.bii.a-star.edu.sg/ (accessed on 1 June 2022)), a popular tool hosted at GISAID that can allow users to link candidate mutations with literature-reported and 3D structurally relevant phenotypes based on information submitted to GISAID. Taken together, MADE provides a swift and powerful tool for the research community to select the best candidate strains without strong signals of egg passage adaptation.

The analysis of adaptive distance provides a unique opportunity connecting adaptive evolution, immunogenicity of the viral strains and vaccine efficacy. For example, in addition to A/Victoria/361/2011 mentioned earlier, in the strain selected for both 2016–2017 and 2017–2018 seasons (i.e., A/Hong Kong/4801/2014), egg passage adaptation generated a series of adaptive substitutions including HA1 T160K, L194P and N96S [27], with the L194P mutation being one of the strongest selected alleles with an ES of 46.6. Human sera collected from individuals vaccinated with egg-based strains displayed much reduced inhibition abilities against circulating strains, leading to one of the most severe influenza seasons since the A/H1N1 pandemic of 2009 in the United States [27]. In addition, even though the adaptive distance calculated based on ES is not a direct measurement of the antigenic property, it behaves very similarly to the antigenic space generated using the hemagglutination inhibition (HI) assay [3]. For example, the PCA map based on ES scores is similar to the antigenic space with discrete islands representing different passage-related substitutions across years. Substitutions with higher enrichment scores tend to be those with large adaptive distance as well as strong antigenic jumps. Thus, the evolutionary analysis of egg passage adaptation provides an important means connecting egg passage adaptation, immunogenicity of the viral strains in humans as well as vaccine effectiveness [28].

It is worth pointing out that in addition to egg adaptation, many other viral [9] and host factors [11,12,13] have also been implicated in the variable vaccine efficacy [4]. For example, using a recently developed method for predicting vaccine efficacy based on antigenic distance [29], we observed that a slightly different trend of VE was predicted across the years (Figure 1D and Appendix A), suggesting that different factors can influence VE through different mechanisms. Even though MADE provides an important computational tool integrating signal of egg passage adaptation, it is still an inference method based on historical data. As influenza viruses are constantly co-evolving with humans, we can imagine egg passage adaptation will also change accordingly through time and it will be important to update the inference procedure continuously. Moreover, further experimental studies will be needed to further confirm functional consequences of egg passage substitutions. Taken together, how to holistically integrate multiple factors and how we can combine computational and experimental evidence for vaccine efficacy will be a crucial question for the field.

Even though many efforts including a universal influenza vaccine [30], non-egg cell lines [31,32], genetically engineered viruses with egg-adapted neuraminidase (NA) [33,34] and DNA vaccines [4] have been explored to improve vaccine efficacy, the transition to newer technologies [35] will likely progress rather slowly due to economic and technological limitations. It is likely that the egg-based vaccine production system will continue for quite some time [4,7]. Given the steadily increasing intensity of egg passage adaptation of the H3N2 influenza strains [18], MADE provides a timely and powerful method for the research and public health community circumventing the impact of egg passage adaptation when selecting seasonal A/H3N2 influenza vaccine strains.

## 5. Conclusions

We have developed a computational package called MADE that can predict VE_ad_ for any candidate vaccine strain. We believe that MADE will help the community select more reliable A/H3N2 influenza strains with minimum egg-passaged adaptation, providing a unique opportunity for improving influenza vaccines.

## Figures and Tables

**Figure 1 vaccines-10-00907-f001:**
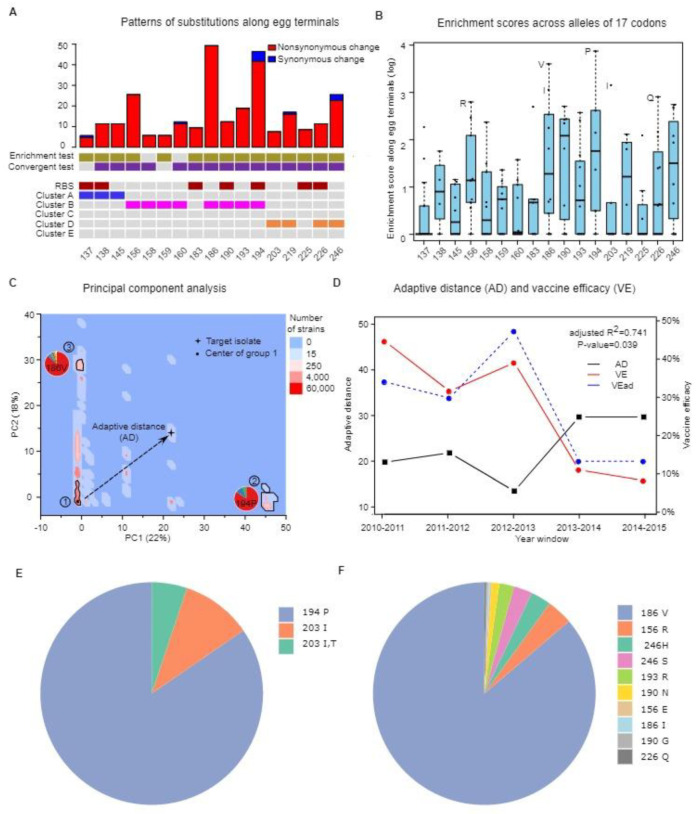
Statistical approaches for characterizing egg passage adaptation. (**A**) Numbers of nonsynonymous and synonymous changes at HA1 codons responsible for egg passage adaptation. The codons showing statistical significance from the enrichment and convergent tests (*q*-value < 0.05) are labelled in yellow and purple, respectively. Codons located in functional domains such as receptor binding sites (RBS), antigenic epitope A, B and D will be labeled in red, blue, violet and orange color respectively. (**B**) Enrichment scores across codons responsible for the egg passage adaptation. Alleles with enrichment scores higher than 20 are labeled. (**C**) Principal component analysis of the 17-dimensional space of ES scores for all the sequences. Discrete subgroups of sequences carrying different passage-related alleles distribute in clusters away from the major cluster. Pie charts display major adaptive alleles in different groups. For example, 194P and 186V are the dominant adaptive alleles observed in cluster 2 and 3, respectively. Adaptive distance (AD) is defined as the distance between the target strain (e.g., CVV) and the centroid of the major cluster for most of the non-egg sequences (i.e., group 1). (**D**) Correlation between the adaptive distance (AD) and vaccine efficacy (VE) between influenza seasons from 2010 to 2015. The predicted VE_ad_ is drawn as the dashed line. (**E**) Proportion of different alleles with enrichment score >10 in the cluster 2 of the PCA map (panel (**C**)). (**F**) Proportion of different alleles with enrichment score >10 in the cluster 3 of the PCA map (panel (**C**)).

**Figure 2 vaccines-10-00907-f002:**
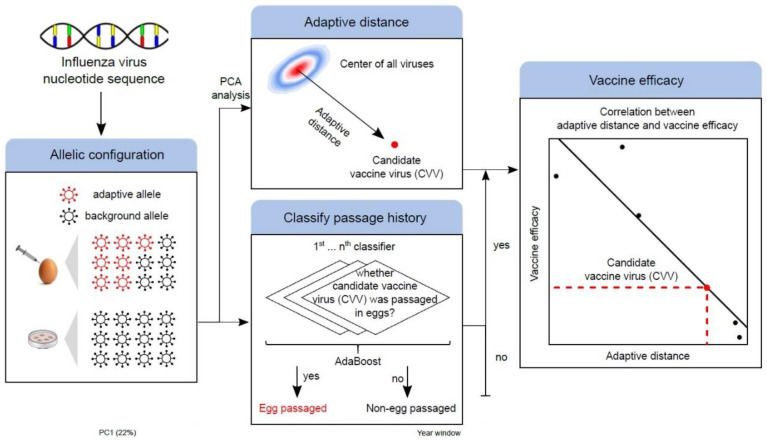
Schematic workflow of MADE. For any candidate vaccine virus (CVV), enrichment scores (ES) of all the alleles (amino acids) at the 17 positively selected HA1 codons were computed. These high dimensional vectors calculated at the 17 codons serve as allelic barcodes for each strain. Principal component analysis of the ES scores for all existing sequences was conducted, and the adaptive distance (AD) for the CVV was computed. Subsequently, MADE predicted the VE_ad_ of the input CVV based on the linear relationship between AD and vaccine effectiveness (VE). In addition, a machine learning algorithm can be applied to classify whether the input sequence with unknown passage history has been grown in embryonated eggs or other passage mediums (e.g., MDCK).

## Data Availability

Two versions of MADE (Measuring Adaptive Distance and vaccine Effectiveness using allelic barcodes) are available at https://github.com/chenh1gis/MADE_docker_v1 (accessed on 31 May 2022) and http://39.105.1.41/made (accessed on 31 May 2022).

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
