# Peer review of "MADE: A Computational Tool for Predicting Vaccine Effectiveness for the Influenza A(H3N2) Virus Adapted to Embryonated Eggs"

_vaccines, 2022, doi:10.3390/vaccines10060907_

Round 1

Reviewer 1 Report

This work draws attention to the problem of a passage adaptation to the egg environment during the vaccine cultivation and proposed (1) the way of computational analysis, which indicated negative correlation between the passage adaptation and vaccine’s efficiency and (2) the computational tool allowing to estimate such an effect operating with an allelic file or a nucleotide sequence.

The manuscript describes the proposed approach in sufficient details and convincing illustrations. The resulting program tool is reported as a docker image in github as well as an online application. The route way of  operating with this program tool is clearly described.

Thus, I suppose that results of this work can be used in practical laboratory studies of vaccines against various strains of influence virus and recommend its accepting for publication.

Author Response

We appreciate the positive and encouraging comments from the reviewer.

Reviewer 2 Report

In general, this work is really interesting and well written. My comments aim to increase the scientific soundness and clarity of it.

  1. Abstract should contain a brief aim of the study (there are background, methods, results and conclusions).
  2. please avoid repetition of the same keywords as used in the title
  3. “vaccine effectiveness” is used for the first time in line 58 and this is the place where it should be abbreviated (not in line 81)
  4. Lines 97-99 actually predict the usefulness of MADE in advance.
  5. Please unify the writing of nonsynonymous to synonymous ratio (line 75 vs. line 188)
  6. Please add a chapter called Conclusions at the end.

Author Response

We appreciate the encouraging comments and constructive remarks from the reviewer. We have followed the reviewer’s advice and revise the manuscript accordingly.  

1) The abstract has been re-organized and a brief aim of the study has been added to the latest version.

2) Updated as suggested by the reviewer.

3) Corrected as suggested by the reviewer.

4) We have modified the wording as suggested.

5) Corrected as suggested by the reviewer.  

6) “Conclusion” section has been added. 

Reviewer 3 Report

1. Authors should rearrange the content from Line no. 10 and 11 of Abstract. Abbreviation MADE was expanded in such a way, that ended with another abbreviation (VEad).

2.  In the 2nd Line of Introduction, reference [1] of WHO is actually dated 6th November, 2018, not 2019 as mentioned in the reference list.

3. In the 2nd paragraph of introduction, please check if “4” is correctly placed in “cacies 4. Among these factors”.

4. In the Abstract, the results shown are that from reference number 9. I got it because the same statement is mentioned in the Introduction section also and there the authors have cited reference 9. How can authors mention a published result [9] as a result of their present research outcome?

5.   In Figure S1, authors should end the legend with “respectively”.

6. In the Subfigure A of Figure S2, Y-axis legend seems missing. Same is for S5 and S6.

7. In Figure S2, authors should mention what represents A, B,C, D, E, RBS, Other?

8. Authors have cited reference 30 after last reference 17, why? Please correct and maintain sequence.

9. Figure 1D and Figure S4 are quite similar.

10. In discussion section, authors should also highlight limitations of the current study.

11. Literature search is quite weak. There was only one cited article from 2020 and nothing from 2021 and 2022.

Author Response

Thank you for your careful review and constructive suggestions, we have followed the reviewer’s advice and have modified the latest version as following:

1) We have updated the abstract as suggested.

2) We have now corrected the reference..

3) 4 represents the reference 4, and we have now updated the citation.

4) The rational of MADE is based on the initial finding from reference 9. In the current work, we have constructed a new tool (MADE) and have updated the dataset to the latest version. We have emphasized this point in the latest text.  

5/6/7/8), we have revised the manuscript as suggested.

9)  Other than MADE, which predict the VE based on the pattern of egg passage adaptation, there is another method which predicts VE based on the substitutions in the viral epitopes. In Figure S4 (updated into new Figure S3), we have compared these two methods. In the latest version, we have re-emphasized this point.

10) We have added a paragraph discussing the limitation of the study.

11) We have added new references following the reviewer’s suggestion.

Reviewer 4 Report

Dear authors

I hope all of you are fine. Regarding the revision of the Manuscript (vaccines-1722947), titled “MADE: a computational tool for predicting vaccine effective-2 ness from the adaptive evolution in embryonated eggs for the 3 influenza H3N2 virus”. Really it is an very important research providing a new tool for vaccine developers to select more reliable strains with minimum egg-passaged changes in the seasonal A/H3N2 influenza vaccine.

However, some comments should be replied.

1-    I advise authors to add Figures S1 and S3 to the main text figures.

2-    In Figure 1A: identify the colors arrowed

3-    Line 305: correct (Figure S4), to (Figure 1D, S4).

Author Response

We thank the reviewer for the encouraging comments.

1) Thank you for the suggestion. We have now moved Figure S3 to the maintext as Figure 1E and 1F. For Figure S1, because it is just a generic phylogenetic tree and is not super informative. We have kept it as a supplementary figure.

2) Thank you for the careful reading. We have updated the figure legends as suggested by the reviewer.

3) We have corrected the text as suggested.

We appreciate your encouraging comments and suggestions in this pandemic era. We are quite fine for now. Thank you!
